# Exploring the path to corruption–An informed grounded theory study on the decision-making process underlying corruption

**Muhammad Untung Manara** [1,2]*, **Annika Nübold**[1], **Suzanne van Gils** [3], **Fred R. H. Zijlstra**[1]

**1** Department of Work and Social Psychology, Maastricht University, Maastricht, Limburg, The Netherlands, **2** Department of Psychology, University of Merdeka Malang, Malang, East Java, Indonesia, **3** Department of Communication and Culture, BI Norwegian Business School, Oslo, Norway

* muhammad.untung@unmer.ac.id

## Abstract

Past corruption research at the individual level has mainly focused on demographics, personality, attitudes, or morality related variables. Until now, only a few studies have focused on the *intra*-individual psychological mechanisms of corruption. Building on normative decision-making theory, the present study attempts to shed further light on the internal mechanisms that lead to the decision that corruption is a viable path. Following an informed grounded theory approach, we conducted semi-structured interviews with 38 Indonesian prisoners who have been convicted of corruption. Guided by a multi-step decision-making process, including problem recognition, information search, and evaluation of the information, our results revealed unique insights into individuals' considerations that led to corruption. We elaborate on interrelations between these stages and explore new forms of corrupt decision-making elements within this process. Theoretical implications for corruption research and the practical implications for anti-corruption programs of these findings are discussed.

## Introduction

Corruption remains one of the biggest and most pressing problems in many countries around the globe. Corruption can be defined as "misuse of an organizational position or authority for personal or organizational (or subunit) gain, where misuse in turn refers to departures from accepted societal norms" [1]. Every year, trillions of dollars—or more than 5% of the global gross domestic product—are lost due to corruption [2]. Moreover, data from the global corruption barometer [3] revealed that one in four people around the world reported that they had to engage in bribery in order to access public services. Thus, the United Nations has identified corruption as the biggest obstacle in their efforts to achieve the 2030 Sustainable Development Goals, which include the eradication of poverty and the improvement of education quality, health, and infrastructure.

Due to its complex nature and its severe impact on organization and society, corruption has been studied by various disciplines such as anthropology, economics, sociology, law,

publicly available would violate the agreement the authors have with the participants. However, data are available upon request from the Research Data Management Team of the Faculty of Psychology and Neuroscience, Maastricht University (contact via datamanagement-fpn@maastrichtuniversity.nl) for researchers who meet the criteria for access to confidential data.

**Funding:** This study was supported by LPDP (Lembaga Pengelola Dana Pendidikan/Indonesian Endowment Fund for Education) 20160822048894. The funders had no role in study design, data collection and analysis, decision to publish, or preparation of the manuscript.

**Competing interests:** The authors have declared that no competing interests exist.

political science, organizational science, and social psychology [4]. In the present study, we are particularly interested in the internal psychological mechanisms leading to corrupt acts and, thus, we will adopt a micro-level perspective to explore drivers of corruption. Specifically, this study looks at corruption from a psychological perspective (i.e., decision-making). Although the corruption literature at the micro-level has explored the effects of both individual and situational factors on corruption, such as personality [5–7], goals and attitudes [8], gender [9], ethical climate [10], ethical leadership [6], and social norms [11–13], there is still scarce research focusing on the cognitive *intra*-individual processes preceding an individual's decision to engage in corruption. This is surprising, since the literature on other unethical behaviors (e.g., cheating, lying, and dishonesty) has provided some evidence that intra-individual cognitive processes, such as intuitive thinking can explain why people engage in these unethical behaviors [14–17]. Furthermore, decision-making processes leading to corruption as a specific sub-form of unethical behavior have, with one exception [8], not been studied yet [18]. Furthering our knowledge in this area is important, however, as research has indicated that corrupt behavior is typically a process that is *not* automatic, but requires thought and consideration [8]. Understanding how individuals think before they engage in corruption is crucial since such knowledge could provide important insights for designing corruption prevention programs. Our results may inform such programs and allow them to take into account what aspects individuals typically consider when they finally choose to engage in corruption.

Furthermore, although insights from initial studies have advanced our understanding of the intra-individual antecedents of unethical and corrupt actions in organizations, these models paid little attention to the *reasons* and *means* that drive the decision to engage in such behaviors in the first place. A better understanding of the underlying cognitive-motivational processes (e.g., goal formation, information processing) would allow us to not only describe, but also explain such decisions, which could then bolster anti-corruption programs. In order to find answers to the questions of *why* and *how* individuals come to the conclusion that corruption is the best way to reach their goals, we particularly aim to explore the internal cognitive-motivational processes underlying corruption. In doing so, we draw on normative decision-making theory [19] and particularly the decision-making model by Engel, Blackwell [20] and use this as a guiding framework.

In the present study, we adopt a qualitative approach using informed grounded theory [21]. While the original version of grounded theory [22] emphasizes pure induction without any prior theoretical knowledge and perceptions, informed grounded theory acknowledges the advantage of pre-existing theories to guide researchers in exploring specific phenomena [21]. This approach is particularly suited for our study because we draw on the decision-making model [20] to guide our exploration of the aspects of decision-making in the data collection and data analysis processes. We will discuss this approach in more detail below.

This study involves a sample of individuals who have been convicted of corruption in Indonesia. Indonesia is an example of a highly corrupt country, as ranked by the Corruption Perception Index (CPI) issued by Transparency International [23]. In this index, low-ranked countries tend to have a high level of corruption, characterized by weak standards of integrity among public officials, a bad judicial system, and little transparency about public expenditure. Our sample is exceptional as there has been little research involving corrupt actors themselves, due to the immoral and illegal nature of corrupt behavior [24]. In the present study, we interviewed 38 imprisoned convicts of corruption, offering us a unique perspective on the intrapersonal thoughts and feelings or behaviors that eventually led to the corrupt action.

In summary, the present study advances the literature on corruption in three important ways. First, this study provides new insights into the psychological mechanisms of corruption by applying a general decision-making framework that goes beyond moral decision-making

theories [e.g., 25, 26] and initial work on decision-making leading to corruption [8]. In this way, we further our understanding of corruption as a rational decision-making process. Second, by adopting a qualitative, informed grounded theory approach [21], the present study answers calls for more diverse approaches in corruption research, such as qualitative interviews with real-life offenders [8]. This is particularly valuable given that most corruption studies to date [8, 11, 27, 28] have entailed lab experiments and did not involve the actors of corruption themselves, thus lacking ecological validity. Finally, shedding light on corrupt behavior by applying a decision-making approach is also valuable from a practical perspective. Understanding the nature of the cognitive-evaluative processes that lead to corrupt behavior enables policymakers to craft interventions that target those key processes more precisely (and potentially all at once), making anti-corruption programs more powerful and effective.

## Theoretical background

**How has ethical/moral decision-making been studied thus far?.** The psychological literature has typically taken a micro-level perspective to study unethical or immoral behavior in organizations. Many ethical decision-making studies [e.g., 29] have focused on understanding the role of two factors for unethical decision-making, often called "bad apples and bad barrels." Bad apples represent individual factors (e.g., cognitive moral development and locus of control), while bad barrels represent organizational factors (e.g., reward systems and outcome expectancy). While this stream of literature has made great contributions regarding the predictive power of individual and contextual factors for moral decisions, it has not explicitly investigated ethical decisions as a dynamic process comprising different cognitive stages.

In contrast, models of moral decision-making, such as the model proposed by Hannah, Avolio [26], usually *do* take into account the psychological processes that are involved in moral actions. The model is based on four psychological mechanisms: moral sensitivity, moral judgment, moral motivation, and moral action. Moral sensitivity refers to the process of identifying the moral problem, interpreting the situation, and identifying various options in order to address the problem. Moral judgment is the process by which the person determines what the most appropriate course of action is. Moral motivation is concerned with the process that increases commitment to a given action. Finally, moral action refers to the decision to engage in a certain behavior in order to address the moral problem. Although Hannah and colleagues' model [26] supports the idea that moral decision-making follows a certain sequence of stages before being translated into behavior, it strongly focuses on the moral content of a situation and the potential reactions to it. The model does not explicitly acknowledge more typical cognitive mechanisms that may drive such judgments and actions (e.g., goal formation or information processing).

Furthermore, although unethical and immoral behavior share conceptual similarities with corruption, they also differ from it in several ways. The concept of unethical behavior subsumes a broad range of behaviors that violate widely accepted (societal) moral norms such as lying, cheating, and stealing [30]. Corruption, which can be seen as one specific form of unethical behavior, additionally includes the misuse of power or authority in an organizational context with far-reaching negative effects, not only on organizations, but society as a whole. The abuse of power is thus essential to distinguishing corruption from other forms of unethical behavior. Although every behavior that violates certain norms has different characteristics and may well follow a different decision-making process [31, 32], scholars have yet to explore whether decision-making in corruption follows the same proposed stages as other forms of norm-violating actions (e.g., unethical and immoral behavior).

As one exception, a study by Rabl and Kühlmann [8] has examined decision- making in the context of corruption. Their proposed model represents a combination of the Model of

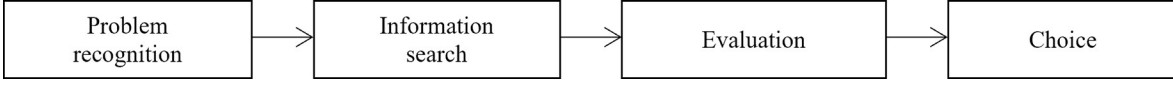

**Fig 1. The stages of the decision making-process (Engel et al., 1986).**

Effortful Decision-Making and Enactment (MEDME) [33] and the Theory of Planned Behavior (TPB) [34]. According to this model, individuals go through two processes before engaging in a corrupt action: firstly, the overall desire and intention to achieve a goal; secondly, the desire and intention to achieve a goal through corrupt action. While this model is valuable for understanding how corrupt action may be initiated by goal striving and intentions, it remains unclear *why* individuals consider corruption to be a suitable means of achieving their goals in the first place (i.e., *how* the decision to engage in corrupt behavior is formed). Thus, we are missing the important link between the intention to achieve a goal and the desire and ultimate decision to achieve the goal through corrupt action is missing.

**Benefits of a general decision-making model to study corruption.** In order to find out how the cognitive-motivational decision-making process leads to corruption, we draw on normative decision-making theory [19] and particularly the decision-making model by Engel, Blackwell [20]. This model describes how individuals make their decisions in a rational way by elaborating on different stages of the decision-making process. This basic model of consumer decision-making has been widely used in marketing research [35]. This model typically explains that there are some stages before consumers chose a specific product: namely, problem recognition, information search, and evaluation. The approach is particularly well suited for shedding light on the questions of *why* and *how* individuals conclude that corruption is the best means for reaching their goals or solving their problems. Thus, it goes beyond current decision-making models of immoral behavior [26] and initial research on corrupt decisions [8]. In contrast to the moral decision-making models discussed above [26, 29], the model proposed by Engel, Blackwell [20] is not limited to particular aspects of decision-making such as moral and specific goals. Thus, the model is more appropriate for the present study because we want to focus on more than just the moral aspects of the decision-making process. To that end, the present study utilizes an informed grounded theory design [21] in order to explore the decision-making process underlying corruption. Using the general decision-making model as a theoretical framework will hopefully reveal new or even contradictory insights. By abstaining from moral aspects, this study may minimize social desirability bias in the data collection process because it can avoid feelings of being judged as participants are not specifically concerned with the question whether the corruption they engaged in is moral or immoral.

The chosen model [20] consists of four stages: problem recognition, information search, evaluation, and choice (see Fig 1). *Problem recognition* involves processes related to identifying and being aware of a problem. The problem is recognized when individuals detect a discrepancy between the current state and a certain desired state [36]. For example, an employee might receive a very low salary (current state), but yearns to buy a house for their family (desired state). The desired state thus becomes a goal that the individual hopes to achieve via a certain behavior [37].

Once the 'problem' and corresponding goal have been identified, individuals move to the second stage: *information search*. This stage includes activities focused on finding potential ways to solve the previously identified problem and reach one's goal. In relation to the previous example, the employee may search for information about the various options available for earning the money to buy a house (e.g., borrowing money from the bank, saving their income, or engaging in corruption). According to Hoyer and MacInnis [35], individuals use a range of

*information sources* when looking for ways to satisfy a certain goal. Sources may include internal sources (e.g., prior knowledge) or external sources (e.g., the internet or colleagues). The *content of information* is the type of information that an individual obtains before deciding to take action: for example, the costs and benefits associated with a particular solution. When individuals try to obtain information, they may focus on only one specific type of information and elaborate on its attributes, or they could search for several alternative pieces of information [38]. In the case of corruption, an individual could search for information about what is actually understood to constitute corruption, what the risks of getting caught are, or what penalties could be expected as a result of engaging in corrupt action. Information search has, to date, received meager attention in the corruption literature or in general research on unethical behavior. For instance, it is not included in the corrupt action model developed by Rabl and Kühlmann [8], nor models of moral/ethical decision-making [26, 29]. However, research from other fields has shown that the source and content of information are both important elements in the decision-making process [36]. Thus, models related to corruption need to incorporate this factor.

Once individuals have gained a certain amount of information on how to achieve their goal, the *evaluation* stage follows. This stage is concerned with processes in which individuals compare and contrast the different options to find out what the best option is [35]. In this stage, individuals examine the information about the attributes of the different options that they have gathered [36]. In relation to our previous example, employees will examine which option is the best for achieving their goal of buying a house; for example, whether borrowing money from the bank, saving money, or engaging in corruption will be the easiest, quickest, or safest option. This stage overlaps with the stage of moral judgment in the literature on moral decision-making [26], which refers to mental processes that determine what action is the most appropriate one to take. However, in those models, little (or no) attention has been given to the underlying reasons behind the conclusion that corruption is the best solution. Exploring the reasons that drove convicts of corruption to see corruption as an adequate solution to their 'problem' is essential for understanding underlying motives behind corrupt action.

The last stage in the process is *choice*. In this stage, individuals choose corruption as the best option among various alternatives. In the corruption literature, there are several classifications of corruption, e.g., individual versus interpersonal corruption [39] or individual versus organizational corruption [18]. Since corruption is a complex phenomenon, there is no universal classification of corruption, however. Examples of corrupt behavior include bribery (giving some form of benefit in exchange for preferential treatment), embezzlement (taking or converting money, property or other valuables of public/organizational funds for personal benefit), and favoritism (misuse of authority to favor family, friends, or one's own party) [40]. Asking individuals convicted of corruption about their concrete actions and thought processes may therefore lead to new, more psychology-oriented insights and potentially the discovery of additional (sub-) forms of corruption.

## Aims of the present study and research questions

Using the normative decision-making model [20] as a framework alongside a qualitative approach (i.e., informed grounded theory) [21], our study explores each stage of the decision-making process for corruption (i.e., problem recognition, information search, evaluation, and choice). We aim to advance the current literature on the decision-making around unethical, immoral, and corrupt behavior [8, 26, 29]. This literature has largely ignored the *reasons* (i.e., goals) and *means* (i.e., information processes and evaluation thereof) that drive individuals' conclusion that corruption is the best option for achieving their personal and professional

goals. Based on the theoretical model by Engel, Blackwell [20], we formulated the following research questions:

*Research question 1: What are the specific goals that individuals convicted of corruption wanted to achieve when engaging in corruption?*

*Research question 2a: What type of information content did individuals convicted of corruption search for before they decided to engage in corruption?*

*Research question 2b: Which sources did individuals convicted of corruption consider when searching for information?*

*Research question 3: What aspects of the different options to act did individuals convicted of corruption consider when they eventually chose corruption as a solution to a specific problem?*

*Research question 4: Which concrete behaviors did individuals convicted of corruption engage in that eventually led to a sentence of corruption and their subsequent imprisonment?*

## Method

### Design

In this qualitative study, we used an informed grounded theory approach [21] to answer the above research questions. Grounded theory is especially appropriate for research topics about which little is yet known [22]. This approach allows researchers to describe phenomena in a detailed way [41] and is especially useful when studying processes [42]. By connecting stages within a process [43], grounded theory allows for the emergence of a new theory and provides insight into the processes between categories. As the decision-making process of corruption has not been investigated yet, and we aim to study decision-making as a process, grounded theory is particularly appropriate for our study.

There are different variants of grounded theory [see for an overview, 44, 45]. In classic grounded theory, the researcher should delay conducting a literature review until the end of the data analysis [22, 46]. The reasoning behind this delay is to keep the researcher free and open to discovering theory from the data and avoiding bias, such as forcing data into a pre-existing theory that may not fit the data. Delaying the literature review in classical grounded theory is based on the ontological assumption of an objective reality [22] which can best be discovered by researchers if they are free from pre-existing knowledge [22, 42]. A popular later version of grounded theory, constructivist grounded theory [47], takes a different, relativist ontological, perspective that assumes that there is no objective reality, but that realities are socially constructed. Therefore, this perspective assumes that individuals cannot avoid their pre-existing knowledge when constructing reality [44]. Building upon this perspective, Thornberg [21] proposed informed grounded theory that is rooted in constructivist grounded theory but particularly emphasizes how a literature review can aid researchers and benefit the grounded theory research process. Thus, according to Thornberg [21], informed grounded theory represents a research process that is grounded in data by grounded theory methods, while being informed by existing research and theoretical frameworks. Rather than considering existing research and theoretical frameworks as obstacles for developing theory from the data, informed grounded theory considers them as sources of inspiration and tools to help a researcher focus on specific aspects and phenomena.

This study adopted an informed grounded theory approach [21] to explore the decision-making process underlying corruption. We draw on the normative decision-making model [20] as a theoretical framework. Therefore, the development of grounded theory in the current study is based on the decision-making model by Engel, Blackwell [20]. Following that model, we identified aspects of four different stages of the decision-making underlying corruption: goal, information search, evaluation, and behavior. In order to identify the aspects of each

stage, we constructed codes, concepts, and theories that were grounded in the data by applying grounded theory methods [21, 41]. We considered the existing corruption literature as important insights to identify new potential concepts or theories in every stage of the decision-making process underlying corruption. Therefore, the development of the decision-making model underlying corruption in this study is based on the decision-making model on newly collected data and considers the existing corruption literature to analyze aspects within the decision-making process.

## Participants

In order to better understand the decision-making process underlying corruption, we studied individuals who had experience with corruption and had been sentenced for their corrupt behavior. Our participants were individuals convicted of corruption in Indonesia. Indonesia's corruption perception index (CPI) scores are consistently below 50 on a scale of 0 (being the most corrupt) to 100 (being the least corrupt) [23] indicating a high incidence of reported corruption cases. Almost every day, the Indonesian media reports about corruption cases, ranging from the regional level to the national level. After the Suharto era, political power in Indonesia became decentralized, leading to a spread of corruption particularly at the regional level [48]. The Corruption Eradication Commission of Indonesia (KPK), for example, reported that at least 32 persons in the position of the head of the regional government were prosecuted and caught in corruption cases between 2015 and 2018 [49]. Thus, because corruption is deeply ingrained into peoples' daily lives, studying corruption in this context is particularly relevant. This study was therefore conducted in three regional prisons in Indonesia. Two of the prisons are male prisons, and one is a female prison. Although access to the specific target group (prisoners convicted of corruption) can be considered extremely challenging (in terms of official approval, but also in terms of openly speaking to and establishing rapport with participants), this study benefitted from one of the authors being from Indonesia, making this process feasible.

To gain access to the prisons, we sought a permit letter to conduct interviews by sending a research proposal to the East Java Regional Office of the Ministry of Justice and Human Rights, Indonesia. When the permit was issued, we brought it to the regional prisons with individuals convicted of corruption. In each prison, the prison authority appointed a public officer to interact with the first author. The first author explained the research proposal, including the characteristics of the study participants. A separate room was provided for interview processes in the prisons, to ensure that conversations would not be overheard by the other prisoners or guards. Each participant who meets the inclusion criteria was called by the public officer to the room to answer the interview questions. The interviews were conducted from May 2017 until August 2017.

The researcher interviewed each participant privately, face to face, using the prison's provided room. Before the interview began, participants were informed about the research including the research context, the purpose of the study, as well as confidentiality of the research process. Then, the written informed consent was signed by each participant who agreed to participate in this study. Participation in this study was fully voluntary. Even though the invited participants were in prison, they had the option to refuse to participate in the interview without any negative consequences. In this study, three invited individuals declined to participate in the interview process.

In grounded theory, the principle of 'theoretical saturation' is used to justify the sample size. Theoretical saturation refers to the point in the data collection and coding process at which no additional new conceptual categories emerge [22]. Throughout the 38 interviews that we collected in three prisons in Indonesia, our coding dictionary stabilized further and

further (while we continuously acknowledged the criteria of conceptual depth suggested by Nelson [50], until we felt that, within the last interviews, a point of data saturation was reached. The participants included 27 men and 11 women who were between 32 and 73 years old ($M$ = 51.7, $SD$ = 9.4) at the time of the interview. Most participants were educated individuals (i.e., 15 participants had a bachelor degree, 13 held a master degree, three were doctors, seven had a senior high school degree, and the rest had obtained another educational level). In terms of organizational employment, 26 participants had worked for public organizations, while 12 had been employed in private organizations. Participants held a variety of positions at the time that they had engaged in corruption (e.g., principal, company owner, lecturer, regional government head, secretary, treasurer, broker, and tax officer). Furthermore, 22 participants had been in a leadership position, while 16 had been subordinates without any supervisory responsibility.

## Data collection

We employed semi-structured interviews to explore the specific decision-making stages that individuals engaged in before deciding to behave corruptly. The interview guide was developed based on the stages of the normative decision-making process [20] described above. We attempted to find information on the four main stages of this process. In line with grounded theory, the interview questions were changed and adapted during the process of interviewing based on the insights gained from the previous interviews [22]. For example, the question about the goal that participants wanted to achieve was revised from "What was your goal when you made that decision?" to "What was the benefit for you of carrying out that behavior?" in order to maximize insights into participants' goals for engaging in corrupt behavior (see S1 Appendix for examples of questions in the interview guide).

All interviews were conducted by the first author, who is a native Indonesian. Having an interviewer who is the same nationality of the interviewees is beneficial for helping participants feel comfortable and allowing them to talk in their native language [51, 52]. Likewise, the shared background can incline participants to feel greater trust toward the interviewer, which is crucial when talking about their unethical behaviors [51, 52].

Because corruption is very delicate topic, we tried to minimize any types of undesirable treatment effects and participant reactions [51, 52]. To this end, we started the interview by expressing empathy for their situation. We began the interview by asking participants about their behavior that led them to prison instead of using the term corruption in order to minimize social desirability bias. To create a safe environment for participants where they could honestly and openly discuss their experiences, we carefully ensured their confidentiality and privacy during the process of data collection. Participants were encouraged to talk about their experiences related to the behavior for which they had been convicted as openly as possible. Most of them were enthusiastic about participating in the interview. They were thankful for being heard and enjoyed being able to talk about their experiences, their coping mechanisms with the prison situation, and their personal opinion about the court decisions with a researcher. After the interview, participants completed a demographic questionnaire. All responses to the interview questions were recorded with an audio recorder (average duration was 45 minutes per participant). The internal ethical review committee at the authors' home university in Europe approved the procedure of this study.

## Data analysis

Following Wilhelmy and colleagues [53], we transcribed the interview data until nearly reaching saturation (i.e., until the number of new categories decreased significantly). Thus, we

transcribed 23 interviews; the remaining 15 interviews were coded directly from the audio files. As suggested by Urquhart [43], the data obtained from the interviews was coded in the original language (Indonesian). It has been recommended that researchers use the original language as far along in the analysis process as possible, in order to capture the experiences of participants in an unbiased way and avoid loss of meaning [54]. Thus, we did not translate the complete interviews; for illustrative purposes, we only translated the codes and the corresponding excerpts from the transcripts. The coding was conducted by two coders (i.e., the first author and a research assistant) who are native Indonesians with excellent English skills. The first author trained the research assistant in three one-hour sessions. This training included how to assign a code to the text and organize categories. Since not all authors had mastered the Indonesian language, our team discussions revolved around the material (codes and excerpts) that was translated into English.

For the data analysis, we followed the three steps specified by grounded theory: open coding, axial coding, and theoretical coding [41]. Coding is a process of deriving and developing concepts from the data at hand [41]. Coding can be done word by word, phrase by phrase, sentence by sentence, or paragraph by paragraph [41]. Following Wilhelmy and colleagues [53], we coded each possible element that we considered worthy of coding, that is, single words, sentences, or whole paragraphs. Constant comparative analysis, which is the analytic process of comparing different pieces of data and looking for similarities and differences [41] occurred in the three coding steps. The constant comparative analysis was conducted based on the informed grounded theory approach [21]. In this process, we considered the existing literature and how this could be used to identify and label new categories [55]. Through all the coding processes, we made use of the coding software MAXQDA 2018.

First, in the *open coding* stage, we analyzed and coded the raw data. The purpose of this coding step is to understand the essence of what is being expressed in the raw data and assign a conceptual name (code) to describe that understanding [41]. Following the procedure described by Corbin and Strauss [41], the two coders independently coded the data and then met to compare and discuss the differences in their individual coding. Following previous grounded theory studies [53, 56, 57], we used a coding dictionary to facilitate the coding process. The coding dictionary is an evolving system of categories that is continually modified (e.g., new codes are added; some codes are changed) based on constant comparison between new codes and existing codes [56]. The two coders recorded their consensus on the appropriate use of code in the coding dictionary.

Second, in the *axial coding* stage, we organized codes into categories in order to elevate them to a more abstract level that is relevant for the research questions [43]. The purpose of this process is to find higher-level concepts called *themes* [41]. We constantly compared the codes to codes that had already been classified into categories or subcategories based on their similarities and differences. For example, all codes related to corrupt behavior could be categorized using the main code *behavior*, which covered possible subcategories such as bribery, embezzlement, favoritism, and manipulation of information. At this stage, the two coders also met to discuss differences in their reasoning for classifying sub-codes into main codes. These categorizations were documented in the coding dictionary.

The third and final step is *theoretical coding*. In this stage, the goal is to link various categories to a core category and reveal an underlying theory [41, 43]. A core category is a conceptual idea that could cover all other categories and represent the core theme of the research topic [41]. In order to investigate the relationships between our categories, we compared categories to each other and discussed the links between them [43]. In this process, we tried to identify categories that occurred together across each stage of the decision-making process. For example, we tried to discover whether any specific goal or information search activity was related to

a certain type of corrupt behavior. To reveal these relationships, we used the *code relations browser* tool of the MAXQDA 2018. This tool is able to identify the relationships between codes by examining codes that were reported together by the participants. Finally, we integrated our findings, identified core categories and links between categories, and developed a diagram that illustrates our emerging theory, grounded in the data [41].

# Results

## Overview

The aim of this study was to shed further light on the decision-making processes of individuals who had been sentenced for corruption. Following the normative perspective on decision-making [20], we first explored the last stage of the decision-making process, that is, participants' behaviors that had been identified as corrupt and for which they had been sentenced. We then proceeded with the first three stages: the goals that our participants wanted to achieve by engaging in corruption; the information that they searched for before deciding to engage in corruption, and the aspects that they considered when choosing corruption as a solution to a certain problem. We chose this specific order, as we first wanted to familiarize participants with the interview setting and the topic [58] and let them explain their point of view before asking more detailed questions about their underlying motives and considerations. Fig 2 presents an overview of our findings. In Table 1, we provide more detailed information about higher-level categories (axial codes), and lower-level categories (open codes).

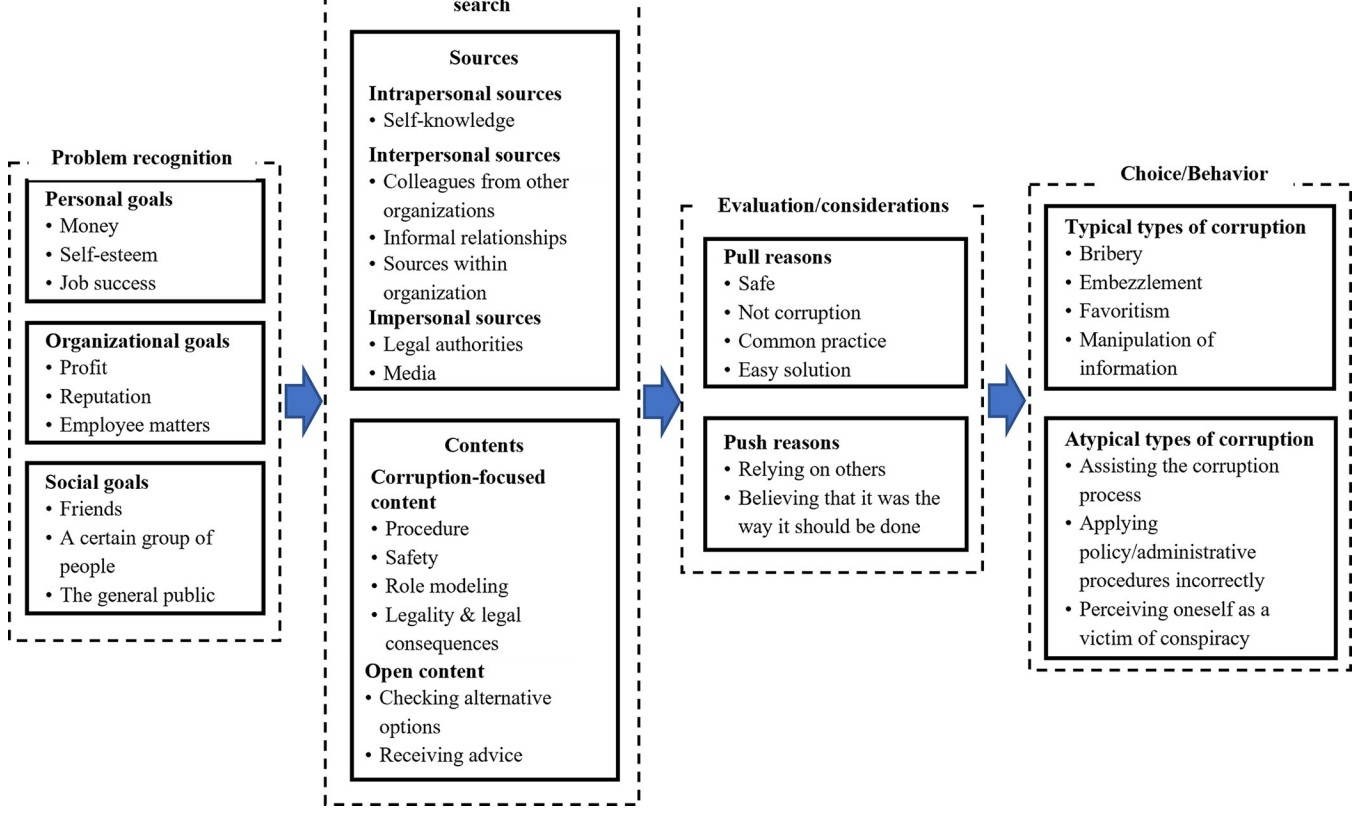

**Fig 2. Conceptual model of decision making-process underlying corruption.**

**Table 1. Axial code, open code, and example quote.**

| Behaviors: Which of participants' behaviors were judged as being corrupt? | |
|---|---|
| **Axial code and open code** | **Example quote** |
| Typical types of corruption | |
| *Embezzlement*: Obtaining money from a project, taking money from the government budget, asking for higher prices | "I took it [the money] all. They did not know that. They only see the data. . . I gathered it from the individuals' taxes. Then I made the report." (Participant 10, a tax officer) |
| *Bribery*: Giving a monetary bribe in order to secure a project, a grant, or lower taxes | "I was only coordinating things so that they would receive the money. The thing was, in the circle of the Ministry of A, if we wanted to secure a project, we would have to be bold, give some [money] to the people in the head office. So, every budget should have their [the ministry's] approval." (Participant 6, a principal) |
| *Favoritism*: Giving a loan from the organizational budget to the football club, to ineligible people, or to small enterprises | "I gave it [the loan], but doing that led to my involvement in this corruption case. I gave it to individuals who were not eligible, to the sub-district head, local police head, and village head. It said it was forbidden in the instruction manual." (Participant 33, a manager of an empowerment program) |
| *Manipulation of information*: Falsifying a document, data, and financial report | "And I, as the treasurer, should be smart, so that we would get the funding again. . . I falsified [the financial report], so that [we] would get the funding again." (Participant 32, a treasurer of an empowerment program) |
| Atypical types of corruption | |
| *Assisting the corruption process*: Assisting in money transfer processes, helping in land acquisition processes, lending the name of one's company to a corrupt project, managing events within the program that is used for one's corupt actions, finding an institution as a partner for the corruption process, and signing a document | "At that time, my company name was borrowed by a colleague of my relative for a uniform procurement project. . . I just put the sign and stamp on it, and they did the job. . . The project was executed well, and I got the fees. Two months later, I was called by the prosecutor. There was a problem (with the project)." (Participant 12, an owner of a company) |
| *Applying policy/administrative procedure incorrectly*: Building a market place in disputed land, signing a contract or receipt for a corrupt project, giving money to a project without obtaining a proper receipt, spending money that was not included in the budget, letting the project grant lapse, acquiring land without a propper appraisal process, investing money into businesses outside the organization's vision, giving a credit without proper management, and applying the wrong procedure when borrowing money from the bank | "I was sentenced because [I] did not use appraisal [in the land acquisition]. However, we had asked the tax office about the appraisal process, but they never processed it. Then, we had a consensus meeting [to determine the price of the land]." (Participant 24, a district head) |
| | "We, as managers, were considered to be violating the procedure, spending the money was not in accordance with the budget plan." (Participant 38, a manager of regional government company) |
| *Perceiving oneself to be a victim of conspiracy*: Selling land to the state company, providing a loan, assisting the farm community in order to increase productivity, and buying the land for the sugar factory | "I was convicted for violating the law, but in reality, it was not like that. . . In short, I was hindered in terms of participating in the political election [regional senator election]. I was tricked with the Corporate Social Responsibility [CSR] program. I managed the CSR program from a public company. . .Finally, political opponents accused the program of corruption, and made me go to jail." (Participant 9, a secretary of farmer organization) |
| **Problems: What are the goals that the participant wanted to achieve with his/her decision to engage in corruption?** | |
| **Axial code and open code** | **Example quote** |
| *Personal goals*: Money (consumptive needs, paying off a debt, buying a home and land, paying for family needs), self-esteem (demonstrating one's ability, being appreciated and praised, being popular), and job-related (following one's job description, career development, getting jobs/projects) | "The point is, what I wanted, was to get the acknowledgement that I was able to do it, able to do a good job." (Participant 32, a treasurer of the empowerment program) |
| | "If I did not follow (the instruction), I would not have gotten the project." (Participants 16, a lecturer) |

*(Continued)*

**Table 1.** (Continued)

| | |
|---|---|
| *Organizational goals*: Supporting a regional government project, organizational profit, organizational reputation, employees' benefits, and helping the organization to get the project | "But it [giving a bribe] was for the common good. For the school's progress (Participant 6, a principal). |
| | "I did it (taking the money) for the empoyees' sake... So, I shared the money, it was not for my own and my family's benefit."(Participant 21, a branch manager of a national company) |
| *Social goals*: Helping the farmers, helping a friend, for public facilities, for the people, and for one's village | "What I truly wanted was to make the road comfortable. [So] Cars can pass through. I just wanted to get funding for the construction, wherever the money comes from." (Participant 11, a community secretary) |

| Information search: Where did the participant search for information? | |
|---|---|
| **Axial code and open code** | **Example quote** |
| Intrapersonal sources | |
| *Self-knowledge*: Previous experiences and knowledge | "No (I didn't search for further information), I had experience working as an accountant. [I had] experience to apply." (Participant 32, a treasurer of the empowerment program) |
| Interpersonal sources | |
| *Colleagues from other organizations*: The principal from another school, a colleague from another district, colleagues from other universities, another company owner, and another private school owner | "Then I asked my friend, who was in the same position, 'Does it occur in your district?', 'never', he said." (Participant 10, a tax officer) |
| *Informal relationships*: Wife and relatives | "I thought about it for so long, sir. Every night, almost every night I talked to my wife about the same issue [whether I should take the money or not]." (Participant 10, a tax officer) |
| *Sources within organization*: Supervisor/leader, board representative, and staff members | "I always coordinated with my treasurer. [She] agreed and I did it. Suppose that she did not agree, it would not have been like this." (Participant 25, a treasurer) |
| Impersonal sources | |
| *Legal authorities*: Government regulations, criminal law books, the law department within the organization, the audit board, the provincial government, a befriended law student, village land database book, and national land agency | "I read the criminal law book and I knew the maximum sentence was six years." (Participant 27, a civil servant) |
| | "The government regulations said that it can be done by achieving consensus." (Participant 24, a district head) |
| *Media*: News and books | "Beforehand, I was, 'huh'! I had read all the books about corruption; I've read the news. If I got arrested, I would have to spend this much time [in jail]." (Participant 10, a tax officer) |

| Information search: What kind of information did the participants search for before they decided to engage in corruption? | |
|---|---|
| **Axial code and open code** | **Example quote** |
| Corruption-focused content | |
| *Procedure*: Checking how to do corruption securely, how to falsify the report, how to keep one's assets safe from corruption, and knowing the procedure for getting a loan | "Yeah, we learned [from others], we were a new district. I learned it [to falsify the report] from the more senior ones." (Participant 32, a treasurer of the empowerment program) |
| *Safety*: Checking whether the behavior would have negative consequences, and whether the chances of getting caught would be high or low | "Yeah, about that, whether this project was safe or not. Many (people) said that it was safe. Everyone said this was safe. This was the governor's program and not a fake program." (Participant 16, a lecturer) |
| *Role modelling (comparison)*: Checking whether other people also did it, whether others' behavior was 'safe', checking how others behaved in the same situation, and if their behavior was common | "At that time, I made a comparison with the other districts related to this program. I saw that they were fine." (Participant 5, a civil servant) |
| | "There were several [schools]. I knew the other schools did it [bribery] as well." (Participant 6, a principal) |

(*Continued*)

**Table 1.** (Continued)

| | |
|---|---|
| *Legality*: Checking whether the behavior violates laws/regulations, whether the behavior is defined as corruption, what the legal basis for the behavior is, and what the status of the land is | "We sent a letter to the province tax department [the appraisal team], but [we] didn't get an answer. Then [we saw] in the presidential decree it said that it could be done by achieving a consensus. . . Finally, we did it, [because] there was a regulation about that." (Participant 24, a district head) |
| *Legal consequences*: Checking the amount of years for the particular sentence | "Before that, I was, 'huh!' I've read all the books about corruption; I've read the news. "If I got arrested, I would have to spend this much time [in jail]." (Participant 10, a tax officer) |
| Open content | |
| *Checking alternative options* | "It was quite long, Sir, I applied for it [the funding] in 2010. . . I started to apply for funding in 2010. It was rejected because I didn't want that [to bribe]." (Participant 6, a principal). |
| | "So, I gave the information about the project to the Dean, and I asked 'if there was an offer like this [a project], whether we should take it or not?'" (Participant 16, a lecturer) |
| *Receiving advice* | "I always coordinated with my treasurer. [She] agreed and I did it. Suppose that she did not agree, it would not have been like this." (Participant 25, a treasurer) |

| Evaluation/consideration: What aspects did the participant consider when he/she chose corruption as a solution to a specific problem? | |
|---|---|
| **Axial code and open code** | **Example quote** |
| Pull reason | |
| *Safe*: The idea that one's own behavior was safe, that *others'* behavior was safe, the notion that one could handle the risks, that the behavior would not be noticed, and that the behavior would not be a problem | "In my organization, many other individuals did it [took the money] for their own sake, but nobody went to the jail. It encouraged me; nobody got caught. Maybe many individuals took more than me, the director maybe, but all of them were free." (Participant 21, a branch manager of a national company) |
| *Not corruption*: The notion that the behavior was not violating any regulations, was based on regulations, that the behavior was auditable, that it *was* the right way of doing things, not receiving the money oneself, not engaging in any corruption oneself, and not understanding the laws related to corruption | "It didn't violate the regulation because the money went to the organizational account. I didn't get any money, and the head of the regional government didn't either. Because of a political conflict, it became a problem [corruption case]." (Participant 35, a secretary of the district government) |
| *Common practice*: Other individuals also did it, it was a *tradition* | "I thought the tradition was to do that [give the bribe]. Almost everyone, also other colleagues, did that as well." (Participant 6, a principal) |
| *Easy solution*: Less complicated, less time consuming, less *expensive*, not having to do anything to get money | "First of all, it was a fast way. APBD [the procedure with the government budget] could have lasted two years. Then, the efficiency was clear. We just needed to make a simple report, less complicated." (Participant 38, a manager of regional government company) |
| Push reason | |
| *Relying on others*: Following instructions, trusting others, behavior was approved by others (leader/committee), and the notion that others would be held responsible for their own (corrupt) behavior | "It was on the Dean's instruction, I did not initiate that. I only followed the instruction. . . So, I did it because of the Dean's instruction." (Participant 16, a lecturer) |
| *Believing that it was the way it should be done*: The idea that nothing is for free, it was impossible without bribery, it was the best way, and there was no other way | "It was what I could do. It was a good way. Want it or not, there were no other ways. It was the way. I was sure about that." (Participant 32, a treasurer of the empowerment program) |

In relation to behavior, we found seven categories that we then organized into two broad categories. The first covers behaviors that represent typical types of corruption in the literature (i.e., embezzlement, bribery, manipulation of information, and favoritism). The second is behaviors that are not generally considered to be typical examples of corruption that we will call atypical types of corruption (i.e., behaviors related to assisting the corruption process, behaviors associated with the false application of policy/administrative procedure, and participants' perception of being the victim of a conspiracy). Regarding participants' goals, the data analysis revealed three different categories: personal goals, organizational goals, and social goals. In terms of the information search stage, we explored two types of information content that participants searched for before deciding to engage in corrupt behavior: corruption-focused and open content. Corruption-focused content is a category of information that only focuses on the attributes of corruption as a behavioral option, whereas open content is a category that focuses on alternative options. Participants searched for information by considering several information sources, namely: intrapersonal sources, interpersonal sources, and impersonal sources. Finally, participants chose corruption as a solution to a specific problem because of two main reasons, which we labeled *pull* and *push* reasons. *Pull* reasons include decisions to participate in corruption that were motivated by a positive or normative evaluation of the corrupt behavior itself, e.g., not considering the behavior to be corrupt, or considering the behavior to be 'safe', an easy solution to the problem, or as accepted and enacted by others. On the other hand, *push* reasons referred to participants' decisions to participate in corrupt behavior because others (e.g., authorities) had involved them in the process of corruption or because their corrupt behavior was considered to be the only solution to a problem.

Considering that corruption is a complex phenomenon and takes on various forms in different contexts [59], we describe our findings on the decision-making process based on the category of corrupt behavior that we found. For every corruption category, we describe the specific behavior and the associated decision-making stages (i.e., goals, information search, and evaluation aspects).

## Typical types of corruption

This category refers to those typical forms of corruption that have already been identified in the literature. This category includes behaviors like bribery, embezzlement, favoritism, and manipulation of information. Bribery is a behavior that involves offering someone money, services, or other valuables in exchange for preferential treatment [13]. As can be seen in Table 1, the data analysis showed a number of bribery behaviors. For example, a school principal said:

> I was only coordinating them. The thing was, in the circle of the Ministry of A, if we want to get some projects, we have to be bold, giving some [money] to individuals in the head office. So, every budget needs their approval. (Participant 6)

Embezzlement is characterized by taking or converting money, property, or other valuables for personal benefit [40]. For example, a tax officer reported: "*I took it [the money] all. They did not know that. They only knew the data. . . From the individuals' taxes, I gathered it. Then I made the false report*" (Participant 10). In the data analysis, we also explored behaviors categorized as *favoritism*: the misuse of authority to favor certain individuals [60]. An example of behavior that could be categorized as favoritism is giving a loan intended for poor people to ineligible people, i.e., people who are not poor. Finally, *manipulation of information* refers to the intended or unintended abuse of (access to) information, such as cheating, violation of secrecy rules, disregarding the confidentiality of information, or concealing information [60].

For instance, some of our participants reported that they had engaged in behaviors such as falsifying a document, the creditor's identity, or a financial report.

Participants who engaged in typical forms of corruption were mostly 'active' decision makers and, thus, responsible for their behavior. They had different positions in their organization, such as school principal, tax officer, manager and secretary of an empowerment program, village head, and government official. They reported a variety of goals underlying their corrupt behaviors, including personal, organizational, and social goals. The majority of them either explicitly or implicitly reported personal goals underlying their corrupt behaviors. For example, a tax official said that he took much money from the tax account for his personal benefit: "*First of all, honestly, I bought a house for my wife. I bought it for 800 million [rupiah], taken from that account*" (Participant 10). The personal goals were not only related to money, but also to psychological benefits, such as a boost of self-esteem. For instance, a treasurer of the village empowerment program who manipulated the financial report said: "*The point is, what I wanted, was to get the acknowledgment that I was able to do it, to do a good job*" (Participant 32). She also stated that she manipulated the financial report to reach a social goal, namely for the village's sake, as she said: "*I only wanted to get the grant again, so the village gets the grant, that was my intention, even though, maybe, I took the wrong way*" (Participant 32). Besides that, some other participants reported that organizational goals motivated them to engage in corruption, such as goals related to organizational reputation, organizational profit, and employee compensation (e.g., passing on the benefits arising from the corrupt action to the employees).

In terms of the information search process, all participants who engaged in typical forms of corruption made social comparisons before they finally acted corruptly. They typically searched for information about how other people, particularly their colleagues, behaved in the same situation. For example, a school principal who engaged in bribery stated: "*There were several [schools]. I knew the other public schools did it [bribery] as well. I have a colleague. He was the one who told me that the others got the projects in the same way*" (Participant 6). Some of the other participants engaging in typical forms of corruption also searched for information regarding the best procedure for engaging in corruption. In addition, some participants engaging in typical forms of corruption had searched for information on the safety and the legal consequences of their behavior. For instance, a tax officer who engaged in embezzlement told us, "*Beforehand, I was, huh!. . . I have read all the books about corruption, I have read the news. . . how long the sentence would be if I get arrested*" (Participant 10). Besides self-knowledge, informal relationships, and impersonal sources, colleagues from the other organizations were the most common information source who participants in this category consulted for information related to corrupt behavior.

Regarding the evaluation aspects of their decision, participants in this category chose corruption as a solution to a specific problem because of several reasons. The reason most frequently cited by participants was that they had considered the behavior to be safe. They made sure that the behavior would not create any problems for them in the future and would not lead to them getting caught. Participants commonly evaluated the behavior as safe based on their own prior experience or on information they had received from close others. For instance, a manager of an empowerment program who engaged in favoritism reported: "*It was because in the first, second, third, and fourth-year, it was okay; there were no issues. Even in 2007, I got the award, sir. So, I continued to dare to do so*" (Participant 33). A branch manager of a national company who engaged in embezzlement similarly said:

> *In my organization, many other individuals did it [took the money] for their own sake, but nobody went to jail. It encouraged me; nobody got caught. Maybe many individuals took more [money] than me, the director maybe, but all of them were free.* (Participant 21)

Most of these participants also reported additional reasons, such as considering the corrupt behavior as common and enacted by many others, as an easy solution to the problem, or even as the only solution to a problem.

## Atypical types of corruption

**Assisting the corruption process.** One of the behaviors that participants described—assisting the corruption process—is not commonly studied in the literature. This behavior captures participants who reported that they had only contributed in a minor way to the whole corruption process. This category was the most common behavior reported by participants. It includes lending the name of one's company to a corrupt project, providing the signature on a document, assisting the money transfer process, or managing events within the program that had been used for one's corrupt actions (see Table 1). This category of behaviors does not define participants as an active or driving force of corruption, but rather as facilitators who contribute to a certain part of the corruption process. The majority of participants of this category were regular employees in their organization, such as a treasurer, lecturer, teacher, general affairs staff, or a third-party who was involved in the corruption process (e.g., villagers; a construction company director). Some participants reported that, at the time, they were unaware that their behavior had contributed to the corruption process. For example, a university treasurer said:

> *I did not know, I was also not involved in the project team. I was not involved at all. I was just asked by the rector [to issue the money], and of course, I did so because I was the treasurer of the rector. If I would not have done it, it would have meant that I did not do my job right.* (Participant 29)

Some participants who had assisted the corruption process, particularly those who were not aware that their activities related to corrupt actions, trusted others' (e.g., their leader's or colleagues') decisions. Therefore, the evaluative aspects underlying their corrupt behaviors were mainly push reasons, such as following the instructions of others, their leader approving their behaviors, and others being held responsible for their behaviors. For example, a villager who signed the documents (related to corrupt projects) without knowing the details of these documents reported: "*It was an educational institution, I thought [they] were more aware of the details of what's being processed, and I thought there wouldn't be any problem. They would not deceive a villager like me, who had a good intention*" (Participant 2). Logically, participants in this category did not search for much information before they took action.

Other participants in this category engaged in information search before they assisted the corrupt behaviors, especially those who were aware that their behaviors were relevant for the corrupt process. The content of information that participants in this category searched for was similar to those who engaged in typical forms of corruption, such as social comparisons and safety issues. The most considered aspect of these participants was whether their behavior would be safe to do. For example, the head of the institute for research and community service at a private university who was involved in a bribery process said:

> *From 1.75 billion [rupiah], 70% was paid back. It means that what my university used was only 30%. However, the report should sum up to 100%. . . Other [universities] refused [such a project] because the money that should be paid back was too huge. My institution was small, and therefore interested [in the project] to have funding [for research].* (Participants 16)

Before he finally engaged in corrupt behavior, he searched for information among his colleagues on whether this would be a problem or not: "*Yeah [I did search for information], whether it was safe or not. Many said it was safe. It was the governor's program*" (Participants 16). He also asked for his manager's approval and compared his situation to other organizations:

*Yes, I think it was quite a lot [of information]. First, I went to the dean, and then, I asked the other universities that had frequently dealt with this kind of project. All of them agreed. . . The fund deductions from the government project were common. All of them understood this.* (Participant 16)

In sum, some participants who assisted or were involved in the corruption process were aware that their activities were corrupt, while others were not. Their decision-making process differed accordingly. The former group engaged in extensive information search before committing to the corrupt process, while the latter largely avoided this stage.

**Applying policy/administrative procedures incorrectly.**   Among the atypical forms of corrupt behaviors, applying policy/administrative procedures incorrectly covers behaviors that deviate from the standard procedures or policy regulations that apply in a certain context. An example of this type of behavior is the acquisition of land without a proper appraisal process. A district head reported:

*I was sentenced because I did not follow the proper appraisal procedure [in the land acquisition]. However, we had asked for an appraisal process at the tax office, but they never processed it. Then, we just had a consensus meeting [to determine the price of the land].* (Participant 24)

Another example stems from a manager of a regional government company who invested organizational money in a way that deviated from the organizational vision: "*I, as a manager, was considered as violating the procedure, spending the money was not according to the budget plan*" (Participant 14). Interestingly, most of the participants in this category were in a position of power in their organization, such as district heads, a managing director of a public company, a village head, a school director, and a school principal. Thus, they had more insights, authority and, thus, opportunities to apply policy and administrative procedures in an incorrect way.

Some participants in this category reported social and organizational goals motivating their behavior, such as improving public facilities and increasing organizational income. However, most of these participants also reported personal goals, including their career, self-esteem, and personal income. These participants mostly searched for information about the legality of the behavior before they finally engaged in the activities that led them to jail. They mostly searched for that information in regulation documents or asked a higher authority, such as the board committee. Therefore, most of the participants in this category did not consider their behavior to be corruption.

**Perceiving oneself as a victim of conspiracy.**   Several participants reported that they were victims of the 'real actor of corruption' and his/her political motives. For instance, a secretary of the farming organization said:

*I was convicted of violating the corruption law, but in reality, it was not like that. . . In short, I was hindered from participating in the political contestation [regional senator election]. I was trapped with the CSR [Corporate Social Responsibility] program. I managed the CSR program*

*for a public company. . . Finally, this program was corrupted based on political motives to make me go to jail.* (Participant 9)

In terms of hierarchy, participants in this category mostly had a high-level position in their organization, including a village head, branch manager of a bank, NGO founder, businessman, and a manager of a regional public company. Participants in this category mostly reported that they had a conflict of interest with people who had more power and higher authority. For instance, a village head told her story:

*I participated [as an incumbent] in the village head election. In that process, my competitor cheated. The regency head appointed this guy [as the elected village head]. . . and I took legal action against the regency head [for his decision] . . . I won the judicial process, and the court asked the regency head to annul the election result. But the regency head appealed to a higher court. However, the higher court rejected his appeal. Finally, the regency head said, "search for any type of mistake that she made [that could be considered violating the law]." I knew this from someone who joined the meeting. . . Then I was sued by the prosecutor for corruption with regard to the building project.* (Participant 30)

These participants did not consider their actions as corrupt behavior or unethical behavior in general. For example, when we asked, "what behavior of yours was regarded as violating the law?", a secretary of a farming organization reported:

*I managed 2.3 billion [rupiah] from a farming company in the form of rice seeds, fertilizers, and the cost of the farm activities. The fund was from the CSR program of a public company. In the end, we failed to harvest because of the poor quality of the seeds, and we were asked to pay back the fund. We could not pay the money back. . .It was not corruption.* (Participant 9)

Regarding the goal formulation, participants in this category did not report any personal interests underlying the behavior that they engaged in. Instead, they reported social goals. For instance, the secretary of a farming organization claimed: "*I did this for social reasons. I did not get any salary for that. I even used my own money for that program*" (Participant 9). Similar to participants who applied policy/administrative incorrectly, these participants believed that their behaviors were not corrupt and did not violate any regulations. For example, a credit analyst in a public bank said: "*We had all of the requirements for that [giving a loan to a businessman]. In terms of regulations, it was not possible to be regarded as violating any regulations*" (Participant 17). Finally, most of these participants did not search for any information, as they had engaged in the respective behavior (which they regarded as legal) many times before.

## Discussion

In the present study, we set out to explore the intra-individual cognitive-motivational decision-making processes underlying corruption. We took an informed grounded theory approach [21] while using a general decision-making model [20] to guide our study. Our findings complement previous models of unethical decision making [e.g., 25, 26] and corrupt action [8]. While these studies have primarily focused on the actor's moral awareness and judgment, or on the ability to solve one's problem with corruption per se, our study focuses on the *why* and *how*, i.e., the cognitive-motivational stages that occur *before* individuals conclude that corruption is the best option to reach their goals. In this way, we further our understanding of corruption as a rational decision-making process. The detailed analysis of our data revealed some interesting new insights. We identified new categories for each stage of the

decision-making process and uncovered previously unconsidered relationships between different aspects of these stages.

Firstly, with regard to corrupt behavior, we found atypical forms of corruption that did not align with the literature's usual categories, like bribery, embezzlement, manipulation of information, and favoritism [40, 60]. These behaviors included applying policy/administrative procedures incorrectly and assisting the corruption process. In addition, some participants perceived themselves to be victims of a conspiracy. In line with the typology of ethical decision outcomes [61], participants engaged in corruption both intentionally and unintentionally.

For these atypical, mostly unintentional forms of corruption, the decision-making process was followed less consistent than what happens with typical forms. According to those participants, they were not (fully) aware that their actions were illegal and could be considered corrupt. Consequently, they reported fewer personal goals or information search activities, and mostly did not actively decide to engage in the behavior that was then later judged as corrupt as it was part of their everyday behavior or because they blindly relied on the judgment of others. For most of the typical forms of corruption, our qualitative data support the idea that the process leading to corruption does indeed resemble general decision-making models and proceeds through different stages, including the identification of a problem and goal formation, information search, and evaluation of this information. Thus, our findings highlight that corruption may involve a more elaborate decision-making process than previously considered in models of unethical and immoral decision-making [26, 29].

Secondly, in the domain of goals, participants not only mentioned their personal and organizational goals [1] but also a number of what we categorized as *social* goals. These included helping farmers, improving public facilities, and aiding the general public. Corruption is generally regarded as immoral behavior [40] used to advance personal and organizational goals [1, 8]. However, our findings indicate that corruption can be a means of achieving pro-social and morally sound goals. This aligns with suggestions by De Graaf and Huberts [62], who proposed that goals like friendship or love, status, and impression management could play a role in corruption.

Thirdly, concerning information search, participants reported both searching for information related to corruption and searching for alternative solutions and advice about whether corruption would be the best solution or not. Participants mainly used interpersonal sources, such as close colleagues, especially from other organizations. This finding emphasizes that corruption is not always performed by an individual in isolation, but often performed by consulting with others [39]. Thus, scholars may need to take a network perspective [4] when studying corruption. Furthermore, we found that participants who reported having had a personal goal related to corruption pursued a more elaborate information search process than participants who reported having had social or organizational goals. Logically, the extent of awareness with regard to the corrupt actions determined the amount of information searched for. Less aware participants reported that they signed a document without reading it in detail, acted based on trust in others, or simply carried out a certain behavior without any further consideration of its consequences. In contrast, more aware participants reported having searched for much more information (e.g., about corruption laws, alternative solutions, and the potentially negative consequences of getting caught). This result suggests that contextual factors (e.g., task type, time pressure, hierarchical structures) can determine whether individuals engage in more or less information search and, thus, rational or intuitive corruption [63].

Finally, in the evaluation stage, we identified what we labeled as push and pull reasons for engaging in corrupt behavior. Push reasons which have not been identified in the literature so far were associated with participants not being aware of the corrupt character of their actions. Participants felt pushed to engage in these behaviors because they trusted and obeyed

authorities (i.e., engaged in corruption because their supervisor involved them in the corruption process). Furthermore, they may have considered corruption as the only solution to a problem, highlighting the role of contextual and systemic pressures in encouraging corrupt decisions. These findings align with the notion that corrupt behavior is influenced by not only individual aspects, but also situational, organizational, and environmental aspects [59, 64, 65]. Pull reasons corresponded to previously reported positive motivations for corruption, such as perceived behavioral control and attitude toward corruption [8], risk of disclosure and the size of the bribe [66], as well as descriptive norms of corruption in or across organizations [11, 67]. Most of the participants that engaged in typical forms of corruption and were aware of their acts used rational cost-and-benefit analysis [4, 68] (e.g., checking whether they would be caught) to reach a decision.

While pull reasons for corruption fit with the notion that bad people intentionally make bad decisions [61], the push reasons listed by our participants suggest that scholars should consider the possibility of unintentional corrupt behavior. While we used rational decision-making theory [20] as a framework for our research, our findings indicated that decision-making in corruption can be both rational and intuitive. This aligns with previous work on unethical behavior and moral decision-making, which has also emphasized rational and intuitive approaches, such as the moral decision-making model [25] versus the literature on intuitive dishonesty [14].

## Theoretical implications and contributions

This study makes several theoretical contributions. First, we contribute to the literature on corrupt decision-making by further exploring the intra-individual, multi-stage process of corruption that goes beyond the previously identified underlying mechanisms and causes for unethical behavior outlined in ethical, moral, and corrupt decision-making models [8, 26, 29]. By exploring additional aspects at every stage of the decision-making process leading to corruption (e.g., *how* do individuals search for information?, *Why* do individuals choose corrupt behavior over another kind of behavior as a solution to a specific problem?) as well as the relationships between them, this research furthers our understanding of the cognitive-motivational mechanisms that lead to corrupt actions. Our study thereby extends previous decision-making models focusing on corruption, such as the corrupt action model [8] which draws on the Theory of Planned Behavior [34]. This model explains that corrupt action is driven by the desire and intention to achieve personal and professional goals through corruption [8]. This desire and intention are affected by several individual factors, including attitudes, subjective norms, and perceived behavioral control [8]. While the corrupt action model [8] also focuses on goals, it does not consider other stages that are typically covered in classical decision making theories. Therefore, our study provides a valuable alternative model for explaining the decision-making process underlying corruption. For example, we explored the information process aspects that were not addressed in the corrupt action model [8] and other ethical decision-making models [26, 29]. We found that individuals searched for information regarding the safety and legality of their actions, and compared how others behaved in the same situation. Furthermore, our findings extend previous studies that focus on personal and organizational causes of corruption [5–7]. This research has for example explored the interaction effect of ethical leadership and followers' dark personality trait Machiavellianism on corruption. This study found that ethical leadership could reduce followers' corruption, particularly when followers' trait Machiavellianism is low [6]. Consistent with those findings, our study found that some participants engaged in corruption because they followed unethical instructions from their leader, indicating the important role leaders may play in followers' corrupt behavior.

Second, using an informed grounded theory approach [21] and interviewing a sample of individuals who had actually been convicted of corruption enriches our understanding of how real-life actors of corruption came to the decision to act corruptly in greater conceptual depth. Most corruption studies have failed to generalize their findings to real-life contexts, as they are mostly conducted in laboratory settings with student samples and using scenarios or games [28]. By investigating the process of corruption with actual convicts of corruption, we were, for example, able to discover atypical forms of corruption (e.g., assisting the corruption process) alongside affirming more typical types (e.g., bribery). We also found that participants engaged in corruption because of push reasons, such as getting involved in corruption by following the instructions of one's supervisor. Thus, the findings of our qualitative approach with real-life actors add to the external validity of corruption research and provide a deeper understanding of the mechanisms involved in making such decisions. Future studies may build on these findings and investigate the corruption process with a stronger systemic perspective (e.g., with social network analysis) that accounts for the dynamic interplay between active and passive actors and the associated intra- and interpersonal mechanisms. Previous corruption studies applying a network approach have indicated that corruption involves multiple actors [69, 70]. Each actor within the network has a different role, engages in different activities, and has individual connections [69]. Analyzing the intra-individual decision-making processes within such a corrupt network may further our understanding of the interplay between intra- and inter-individual factors in corrupt decision-making processes.

## Practical implications and contributions

Our findings may help decision-makers in designing anti-corruption interventions or developing new policies. More specifically, our model provides detailed information about the stages of goal identification, information search, evaluation, and corrupt actions, in addition to the interrelations between these stages. Our results show that the type of corruption the person engaged in determines the decision-making process underlying corrupt behavior. For example, the most frequently reported reason for why participants engaged in bribery was that it was common practice. On the other hand, the most frequently mentioned reason for why participants applied policy/administrative procedures incorrectly was because they did not consider their behavior to be corrupt. Thus, decision-makers should tailor their interventions to the unique aspects of decision-making involved in a specific form of corruption [71], rather than utilize a one-size-fits-all solution.

It is important to note that participants who engaged in typical forms of corruption most often considered the issue of safety. Most of the participants concluded that their behavior was safe and that they would not be caught. They came to this conclusion based on their consultation with others and on their own prior experiences. This finding is consistent with the idea that the ethical climate within an organization plays a crucial role in ethical decision-making [72]. Specifically, individuals are more likely to behave corruptly when they work in a context where unethical behavior (e.g., corruption) is not punished, but is instead a socially accepted norm. Based on these findings, we emphasize the importance of considering the ethical climate when designing intervention programs to reduce or prevent corruption.

Another of our findings that may prove useful for policymakers is that some participants believed that their behavior was not corrupt. This indicates that they did not know which behaviors can and cannot be classified as corrupt. Consequently, we highlight the importance of distributing information about corruption laws in order to increase individuals' awareness and understanding of what is and is not legal. In addition, interventions need to strengthen individuals' personal responsibility for their own actions, their vigilance toward doubtful

supervisory behaviors, and their self-esteem for withstanding orders from authorities that they are critical about. These may be useful complementary strategies in addition to fostering an ethical climate in organizations. Increasing awareness, vigilance, and self-responsibility in individual actors may also help to reduce automatic responding. Following a dual process logic, people could be encouraged to engage in more effortful information processing before proceeding with corruption. This could help to prevent people from unintentionally engaging in corrupt processes due to a restricted decision-making process, as described above.

## Study limitations

Despite the contributions that this study makes, as outlined above, it also has several limitations. One such limitation is that our participants were all from Indonesia. It is possible that the results for individuals from other countries (e.g., Western countries in North America or Europe) will differ because previous studies have shown that corruption in one country is related to cultural aspects [73]. Furthermore, our participants' corrupt behavior was only enacted at a regional level, and we did not investigate corrupt behavior at a national level. Including acts of corruption at a national level might have given us a more comprehensive understanding of large-scale acts of corruption (e.g., grand corruptions). Nevertheless, our sample was still quite diverse and included both men and women who worked in different types of organizations (private/public) and in a variety of positions (e.g., principal, lecturer, and regional government head, secretary, treasurer, and tax officer). In addition to the heterogeneity of our sample, the fact that we collected data until reaching theoretical saturation [22] helped to ensure that our insights may generalize to other samples and contexts.

Another potential limitation of our study design is that participants reported their corrupt behavior retrospectively. Because corruption is a socially undesirable act, it can create cognitive dissonance (i.e., discomfort arising from the idea that one is a good person but has committed a bad act). Therefore, we cannot rule out the possibility that participants engaged in retrospective rationalization in order to reduce cognitive dissonance [74], leading them to reframe their past corrupt behaviors as normal and acceptable [1]. Specifically, the denial of responsibility–one form of rationalization where individuals view circumstances beyond their control as responsible for their corrupt actions [74]–may explain why several participants reported that they only contributed in a small way to the corruption process, were victims of a conspiracy, simply followed orders, or did not have any other choice. In summary, we cannot rule out the possibility that cognitive reappraisal and justification processes influenced participants' answers, such that they did not accurately reflect their original decision-making processes at the time of their corrupt involvement. However, in an attempt to limit this kind of desirability bias, we ensured complete confidentiality during the research process in order to make participants feel safe and allow them to talk honestly about their experiences [75]. Furthermore, as many participants in our sample did report intentional corrupt behaviors and did not deny responsibility, we believe that our findings may rather reflect an empirical reality rather than simply retrospective rationalization. Nonetheless, future studies should examine the decision-making process more directly and with less delay to avoid the issue of retrospection bias (e.g., in an event sampling study, if possible). Finally, we acknowledge that our intrapersonal decision making focused approach may have limited the range of possible underlying mechanisms for corruption that could be identified in our research. Research in adjacent disciplines such as law, social psychology, or sociology could fruitfully employ our method to identify discipline specific mechanisms that could extend the model presented in this research.

## Supporting information

**S1 Appendix. Interview guide.**
(DOCX)

**S1 File. Coding dictionary.**
(PDF)

## Acknowledgments

The authors thank Ika N. Listyanti and Muhammad S. Muntafi for additional assistance.

## Author Contributions

**Conceptualization:** Muhammad Untung Manara, Annika Nübold, Suzanne van Gils, Fred R. H. Zijlstra.

**Data curation:** Muhammad Untung Manara.

**Formal analysis:** Muhammad Untung Manara, Annika Nübold, Suzanne van Gils.

**Investigation:** Muhammad Untung Manara.

**Methodology:** Muhammad Untung Manara, Annika Nübold, Suzanne van Gils.

**Project administration:** Muhammad Untung Manara.

**Supervision:** Annika Nübold, Suzanne van Gils, Fred R. H. Zijlstra.

**Writing – original draft:** Muhammad Untung Manara.

**Writing – review & editing:** Muhammad Untung Manara, Annika Nübold, Suzanne van Gils, Fred R. H. Zijlstra.

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
