## [Decision Letter · Decision Letter 0]

28 Apr 2023

PONE-D-23-05981Exploring the path to corruption – An informed grounded theory study on the decision-making process underlying corruptionPLOS ONE

Dear Dr. Manara,

Thank you for submitting your manuscript to PLOS ONE. After careful consideration, we feel that it has merit but does not fully meet PLOS ONE’s publication criteria as it currently stands. Therefore, we invite you to submit a revised version of the manuscript that addresses the points raised during the review process.

I recommend that it should be revised taking into account the changes requested by the reviewers. Since the requested changes include valuable and constructive reviews, I would like to give you a chance to revise your manuscript. The revised manuscript will undergo the next round of review by same reviewers.

We look forward to receiving your revised manuscript.

Kind regards,

Baogui Xin, Ph.D.

Academic Editor

PLOS ONE

Reviewers' comments:

Reviewer's Responses to Questions

**Comments to the Author**

1. Is the manuscript technically sound, and do the data support the conclusions?

Reviewer #1: Partly

Reviewer #2: Yes

2. Has the statistical analysis been performed appropriately and rigorously? 

Reviewer #1: N/A

Reviewer #2: Yes

3. Have the authors made all data underlying the findings in their manuscript fully available?

Reviewer #1: No

Reviewer #2: Yes

4. Is the manuscript presented in an intelligible fashion and written in standard English?

Reviewer #1: Yes

Reviewer #2: Yes

5. Review Comments to the Author

Reviewer #1: General comment – First, I am pleased to have the opportunity to review this manuscript. This research aims to explore the intra-individual psychological mechanisms of corruption. The topic is very interesting because, in many countries, particularly in developing ones, corruption has been a significant social problem for years. Understanding how corruption occurs contributes to the existing literature and assists decision-makers in designing anti-corruption programs and policies. In general, the authors have successfully identified the research gap, applied the appropriate methodology, conducted solid data analysis, and presented insightful results. However, I believe that the manuscript requires minor revisions before being published in PLOS ONE.

In the following paragraphs, I will list the specific comments on the issues that I have identified after reviewing the manuscript.

Comment 1 – I suggest the authors to use the correct citation format when submitting the manuscript to PLOS ONE. The detailed submission Guidelines can be found at this link: https://journals.plos.org/plosone/s/submission-guidelines.

Comment 2 – The authors have mentioned that the data is not available for download because of some restrictions. I highly recommend that the authors provide the transcriptions of the interviews, as they are excellent endorsement materials to justify the integrity of qualitative research. However, if the data is not available due to some restrictions, such as local laws and regulations, the authors should explain them.

Comment 3 – On p. 3, the authors have mentioned:

“Although the corruption literature at the micro-level has explored the effects of both individual (e.g., personality; 5) and situational factors (e.g., organizational climate; 6) on unethical behavior in organizations, there is still scarce research focusing on the cognitive intra-individual processes preceding an individual's decision to engage in unethical action (e.g., 7).”

I have noticed that only three references are cited. I recommend the authors to enhance the literature review in the introduction section, as this part helps readers to understand the following questions:

1. What is the problem?

2. Why is the problem important?

Without fully understanding these questions, it is hard for readers to comprehend what conflicts or unanswered questions your research addresses.

Comment 4 – On pp. 5-6, the authors have extensively stated the theoretical contributions of their research. From my viewpoint, the authors could further condense and summarise the contributions in the introduction section, leaving more space for explaining them in detail in the discussion section.

Comment 5 – On p.12, the authors have mentioned that there are different variants of grounded theory, which is correctly described. However, I suggest the authors to briefly introduce the characteristics of each main variant of grounded theory, apart from the classical ground theory. For instance, evolved grounded theory (Corbin & Strauss, 2015) and constructivist grounded theory (Charmaz, 2014) are important methodological genres, and each of them is based on different ontological and epistemological assumptions.

Meanwhile, I infer that the approach that the authors implemented (Thornberg, 2012) has some similarities with the constructivist grounded theory (Charmaz, 2014). Therefore, I would like to invite the authors to explain the main differences between the informed grounded theory (Thornberg, 2012) and the constructivist grounded theory (Charmaz, 2014)

Comment 6 – On p.13, the authors have mentioned:

“Our participants were individuals convicted of corruption in Indonesia.”

As a reader, I would like to invite the authors to explain why the convinced informants with Indonesian nationality were chosen in the sampling frame.

Comment 7 – I highly recommend the authors place the section of limitations and future research after the section of practical implications because the current arrangement in the manuscript does not follow the normal logic.

References:

Charmaz, K. (2014). Constructing grounded theory (2nd ed.). Sage.

Corbin, J., & Strauss, A. (2015). Basics of Qualitative Research (4th ed.). SAGE Publications.

Thornberg, R. (2012). Informed Grounded Theory. Scandinavian Journal of Educational Research, 56(3), 243–259. https://doi.org/10.1080/00313831.2011.581686

Reviewer #2: In the manuscript titled ‘Exploring the path to corruption – An informed grounded theory study on the decision-making process underlying corruption’, Authors explore new forms of corrupt decision-making elements within this process. This study contains some interesting findings and are valuable for the understanding of cause for corruption. However, the study still has some flaws. Therefore, MAJOR revision has to be done before this manuscript could be accepted.

（1） As we know, theoretical saturation is very important. The study conducted 38 interviews. Can you tell us whether the theoretical saturation occurred after 38 interviews or in 38 interviews?

（2） The author clearly summarizes the types of corruption, but fails to reveal the underlying causes of corruption. So，it is necessary to further explain the causes of corruption by combining psychological , sociological theories and interview data.

（3）The author puts forward that the goals of corruption includes individual goals, organizational goals and social goals. It will be more meaningful if the author can construct a model of multiple factors leading to corruption through interview data.

6. PLOS authors have the option to publish the peer review history of their article (what does this mean?). If published, this will include your full peer review and any attached files.

Reviewer #1: No

Reviewer #2: No

---

## [Author Response · Author response to Decision Letter 0]

29 Jun 2023

Dear Prof. Baogui Xin,

We would like to thank you for the opportunity to revise and resubmit our manuscript to PLOS ONE, and the valuable feedback we have received. We have substantially revised the manuscript according to your and the reviewers’ comments. In the following, we will present a point by point discussion of how we have dealt with each of the points raised. 

Responses to the Comments of the Editor

Response: Thank you for your reminder to follow PLOS ONE’s style requirements. We have now revised the manuscript following the requirements. 

Response: Thank you for inviting us to provide further details about this issue. We now have mentioned that participants were informed about the study and the written informed consent was signed by each participant (p.14, line 331-334). 

3. In your Data Availability statement, you have not specified where the minimal data set underlying the results described in your manuscript can be found. PLOS defines a study's minimal data set as the underlying data used to reach the conclusions drawn in the manuscript and any additional data required to replicate the reported study findings in their entirety. All PLOS journals require that the minimal data set be made fully available.

Response: Thank you for this suggestion. Corruption is a highly sensitive issue as it is an illegal act. We would like to refrain from making the transcriptions of the interviews publicly accessible as they contain sensitive data such as positions, locations, and types of organizations related to the respective corrupt actions. Although names of individuals and companies have been removed from the transcriptions, readers still can infer and trace people back based on the information from the dialogues in the interview transcriptions. In addition, we informed participants that the research process was treated confidentially including the interview data before they signed the informed consent. This was particularly important for creating trust and rapport, a necessary condition for participants to openly share their experiences. In sum, publicly sharing the full transcriptions does not seem warranted to us as it will violate the agreement with our participants. 

However, we have already provided many excerpts and quotes from the transcriptions which are relevant to the study within the paper (i.e., Table 1). In addition, in case additional data is required by other scholars, for example to replicate reported study findings, the transcriptions of the interviews are available upon request. We now explain this in Data Availability Statement. 

Responses to the Comments of Reviewer #1

General comment – First, I am pleased to have the opportunity to review this manuscript. This research aims to explore the intra-individual psychological mechanisms of corruption. The topic is very interesting because, in many countries, particularly in developing ones, corruption has been a significant social problem for years. Understanding how corruption occurs contributes to the existing literature and assists decision-makers in designing anti-corruption programs and policies. In general, the authors have successfully identified the research gap, applied the appropriate methodology, conducted solid data analysis, and presented insightful results. However, I believe that the manuscript requires minor revisions before being published in PLOS ONE.

In the following paragraphs, I will list the specific comments on the issues that I have identified after reviewing the manuscript.

Response: Thank you for your positive feedback. We found it very motivating to read that you deemed our work interesting and relevant.

Comment 1 – I suggest the authors to use the correct citation format when submitting the manuscript to PLOS ONE. The detailed submission Guidelines can be found at this link: https://journals.plos.org/plosone/s/submission-guidelines.

Response: Thank you for noticing the incorrect citation format. We have now followed the submission guidelines and use the correct citation format. 

Comment 2 – The authors have mentioned that the data is not available for download because of some restrictions. I highly recommend that the authors provide the transcriptions of the interviews, as they are excellent endorsement materials to justify the integrity of qualitative research. However, if the data is not available due to some restrictions, such as local laws and regulations, the authors should explain them.

Response: Thank you for this suggestion. As mentioned in our response to the editor’s comment 3 of the journal requirements above, corruption is a highly sensitive issue and the transcriptions contain sensitive data which potentially allow identifying our participants. We therefore would like to refrain from making our transcripts publicly available. In addition, publicly sharing the transcriptions would violate the agreement we have with participants. We now explain this on Data Availability Statement in the submission system and point to the option that transcriptions of interviews are available upon request. 

Comment 3 – On p. 3, the authors have mentioned:

“Although the corruption literature at the micro-level has explored the effects of both individual (e.g., personality; 5) and situational factors (e.g., organizational climate; 6) on unethical behavior in organizations, there is still scarce research focusing on the cognitive intra-individual processes preceding an individual's decision to engage in unethical action (e.g., 7).”

I have noticed that only three references are cited. I recommend the authors to enhance the literature review in the introduction section, as this part helps readers to understand the following questions:

1. What is the problem?

2. Why is the problem important?

Without fully understanding these questions, it is hard for readers to comprehend what conflicts or unanswered questions your research addresses. 

Response: Thank you for the suggestion. We have extended our literature review considerably and now cite more references regarding the micro-level predictors of corruption (p. 3, lines 55-57). Furthermore, we also added further explanations as to highlight the research gap (i.e., problematize the literature) and explain why it is important to close that gap. For example, we now explain that while the literature on other forms of unethical behavior has emphasized cognitive processes in explaining unethical decisions, the corruption literature has surprisingly not investigated such processes as a mechanism that may explain corruption (p. 3, lines 58-461). Therefore, we want to fill this gap. We also extended why it is important to study such cognitive processes in the last part of the paragraph (lines 65-69) and highlight that our study provides new insights into how and why individuals conclude that corruption is the best option to reach their goals. 

Comment 4 – On pp. 5-6, the authors have extensively stated the theoretical contributions of their research. From my viewpoint, the authors could further condense and summarize the contributions in the introduction section, leaving more space for explaining them in detail in the discussion section.

Response: Thank you for your suggestion. We have now further summarized the theoretical contribution part and condensed it into one paragraph. Furthermore, we have added a more detailed discussion of our contributions in the beginning of the discussion section on page 36, lines 687-693.

Comment 5 – On p.12, the authors have mentioned that there are different variants of grounded theory, which is correctly described. However, I suggest the authors to briefly introduce the characteristics of each main variant of grounded theory, apart from the classical ground theory. For instance, evolved grounded theory (Corbin & Strauss, 2015) and constructivist grounded theory (Charmaz, 2014) are important methodological genres, and each of them is based on different ontological and epistemological assumptions.

Meanwhile, I infer that the approach that the authors implemented (Thornberg, 2012) has some similarities with the constructivist grounded theory (Charmaz, 2014). Therefore, I would like to invite the authors to explain the main differences between the informed grounded theory (Thornberg, 2012) and the constructivist grounded theory (Charmaz, 2014).

Response: Thank you for this suggestion. We now explain that informed grounded theory (Thornberg, 2012) has its root in constructive grounded theory (Charmaz, 2014) but that it adds literature review strategies to the grounded theory research process (p. 12, lines 279-282). In addition, we briefly explain the main differences between classical and constructivist grounded theory (p.12, lines 273-279). Finally, to give readers an overview of other variants of grounded theory, we have now added some references which specifically address that issue (p.12, line 269). 

Comment 6 – On p.13, the authors have mentioned:

“Our participants were individuals convicted of corruption in Indonesia.”

As a reader, I would like to invite the authors to explain why the convinced informants with Indonesian nationality were chosen in the sampling frame.

Response: Thank you for the invitation to explain why we chose Indonesian participants. We now have explained our reasoning on page 13, lines 305-308. Specifically, we explain that because Indonesia scores high on the corruption index and corruption is deeply ingrained into peoples’ daily lives, studying corruption in this context is particularly relevant. Furthermore, we also explained that although access to the specific target group (prisoners convicted of corruption) can be considered extremely challenging (in terms of official approval, but also in terms of openly speaking to and establishing rapport with participants), this study benefitted from one of the authors being from Indonesia, making this process feasible (p. 14, lines 315-219). 

Comment 7 – I highly recommend the authors place the section of limitations and future research after the section of practical implications because the current arrangement in the manuscript does not follow the normal logic.

Response: Thank you for the recommendation. We have now placed the section of study limitations after the section of practical implications. 

Responses to the Comments of Reviewer #2

In the manuscript titled ‘Exploring the path to corruption – An informed grounded theory study on the decision-making process underlying corruption’, Authors explore new forms of corrupt decision-making elements within this process. This study contains some interesting findings and are valuable for the understanding of cause for corruption. However, the study still has some flaws. Therefore, MAJOR revision has to be done before this manuscript could be accepted.

(1） As we know, theoretical saturation is very important. The study conducted 38 interviews. Can you tell us whether the theoretical saturation occurred after 38 interviews or in 38 interviews?

Response: Thank you for this important question and for noticing this issue. Saturation is known as an important but challenging concept in qualitative research. In line with Braun and Clarke (2021), our understanding is that saturation is not about the number of interviews per se, but about the conceptual depth that is reached based on the data. Throughout the 38 interviews, we continuously acknowledged the criteria of conceptual depth suggested by Nelson (2017) to ensure that we reached saturation: (a) range (i.e., are there multiple examples of concepts in the data?), (b) complexity (i.e., is there a rich network of themes and concepts with complex connections?), (c) subtlety (i.e., have instances of same codes been compared to tease out meaning?), (d) resonance (i.e., can the emerging concepts be connected to existing literature in the area being investigated), and (e) validity (i.e., do the findings seem useful for others?). Furthermore, throughout the 38 interviews, our coding dictionary stabilized further and further, and, although it is difficult to pinpoint the exact moment in time, we agreed that within the last interviews, a point was reached when “further data gathering and analyses add little new to the conceptualization” (Corbin & Strauss, 2008, p. 263). Thus, we do agree that saturation was reached in 38 interviews, rather than after 38 interviews. We now explicitly state this in the manuscript on p. 15 (lines 338-344), along with our strategy and understanding of theoretical saturation.

(2） The author clearly summarizes the types of corruption, but fails to reveal the underlying causes of corruption. So, it is necessary to further explain the causes of corruption by combining psychological, sociological theories and interview data.

Response: Thank you for your suggestion. We agree that discussing and further exploring the underlying causes of corruption is an important endeavor. However, the topic of corruption has been discussed in a wide range of disciplines, such as psychology, sociology, or law, as we mention in the introduction and have added to the discussion. As it is not feasible to cover every possible perspective on underlying reasons for corruption, we chose to explicitly focus on ‘a limited set of mechanisms’ of corruption, namely the intra-personal decision-making process underlying corruption based on the steps of a rational decision-making model (Engel et al., 1986). We have indicated this focus more clearly now at various points in the paper (p. 3, line 52; p. 39, line 768-769). In the decision-making process that we reveal through our interviews, we discovered some important cognitive processes that lead up to corrupt actions, such as the goals that drive individuals to engage in corruption, the information sources they consult, as well as aspects that individuals consider when engaging in corruption. In order to meet your request to broaden our perspective, we now added a discussion on how our findings complement previous models that focus on inter-personal and organizational causes of corruption (p. 39-40, lines 774-793). Previous models, such as the corrupt action model (Rabl & Kühlmann, 2008) focused on desire and intention to achieve goals through corruption and some factors (i.e., attitudes, subjective norms, and perceived behavioral control) that affected such desire and intention. Complementing these models, our study found some important additional underlying mechanisms, for example the insight that some participants considered that their behavior was not corrupt. We also cited earlier research that examined an inter-personal cause of corruption (Manara et al., 2020), which examines the interaction effect of ethical leadership and followers’ trait Machiavellianism (i.e., manipulative personality) on corruption. It also shows that ethical leadership has the power to reduce corruption, particularly when followers’ trait Machiavellianism is low. Adding this research may provide insights into the psychological causes and mechanisms already discussed in the literature.

3）The author puts forward that the goals of corruption includes individual goals, organizational goals and social goals. It will be more meaningful if the author can construct a model of multiple factors leading to corruption through interview data.

Response: Thank you for your suggestion. As we have explained in response to your Comment 2, our paper explored the decision-making process underlying corruption following the decision-making stages of the general decision-making model by (Engel et al., 1986). As we explicitly used this model as a theoretical framework, we focused on those stages and found some interesting aspects for every stage. Although we did not specify multiple factors (i.e., additional predictors equivalent to the position of goals in our model) leading to corruption, we explicitly acknowledge multiple factors leading to corruption by identifying aspects of every single stage that could lead people to engage in corruption. Therefore, we do consider multiple ‘causes’ of corruption, but those are of intra-personal nature. For example, in the evaluation stage, we found that participants considered whether their behaviors would be safe or would be punished. Therefore, independent of the initial goals that people specified for themselves, the perception that corrupt behavior would or would not be punished could lead people to engage in corruption or not. We do believe that exploring those cognitive, intra-personal ‘causes’ is equally valid and important as exploring inter-personal or organizational factors which previous models have considered (e.g., Rabl & Kühlmann, 2008; Treviño, 1986; Trevino & Youngblood, 1990). Thus, our study complements such research in a meaningful way. As discussed above, we have also added a statement in the discussion section acknowledging that other factors could be identified by taking another disciplinary lens to the same method.

References

Braun, V., & Clarke, V. (2021). To saturate or not to saturate? Questioning data saturation as a useful concept for thematic analysis and sample-size rationales. Qualitative Research in Sport, Exercise and Health, 13(2), 201-216. https://doi.org/10.1080/2159676X.2019.1704846

Charmaz, K. (2014). Constructing grounded theory (2nd ed. ed.). Sage. 

Corbin, J., & Strauss, A. (2008). Basics of qualitative research (3 ed.). Sage. 

Engel, J. F., Blackwell, R. D., & Miniard, P. W. (1986). Consumer behavior (5 ed.). Dryden. 

Manara, M. U., van Gils, S., Nübold, A., & Zijlstra, F. R. H. (2020). Corruption, fast or slow? Ethical leadership interacts with Machiavellianism to influence intuitive thinking and corruption. Frontiers in Psychology, 11, Article 578419. https://doi.org/10.3389/fpsyg.2020.578419

Nelson, J. (2017). Using conceptual depth criteria: addressing the challenge of reaching saturation in qualitative research. Qualitative Research, 17(5), 554-570. https://doi.org/10.1177/1468794116679873

Rabl, T., & Kühlmann, T. M. (2008). Understanding corruption in organizations – Development and empirical assessment of an action model. Journal of Business Ethics, 82(2), 477-495. https://doi.org/10.1007/s10551-008-9898-6

Thornberg, R. (2012). Informed grounded theory. Scandinavian Journal of Educational Research, 56(3), 243-259. https://doi.org/10.1080/00313831.2011.581686

Treviño, L. K. (1986). Ethical decision making in organizations: A person-situation interactionist model. Academy of Management Review, 11(3), 601-617. https://doi.org/10.5465/amr.1986.4306235

Trevino, L. K., & Youngblood, S. A. (1990). Bad apples in bad barrels: A causal analysis of ethical decision-making behavior. Journal of Applied Psychology, 75(4), 378-385. https://doi.org/10.1037/0021-9010.75.4.378

---

## [Decision Letter · Decision Letter 1]

24 Jul 2023

PONE-D-23-05981R1Exploring the path to corruption – An informed grounded theory study on the decision-making process underlying corruptionPLOS ONE

Dear Dr. Manara,

Thank you for submitting your manuscript to PLOS ONE. After careful consideration, we feel that it has merit but does not fully meet PLOS ONE’s publication criteria as it currently stands. Therefore, we invite you to submit a revised version of the manuscript that addresses the points raised during the review process.

I recommend that it should be revised taking into account the changes requested by Reviewers. I would like to give you a chance to revise your manuscript. To speed the review process, the manuscript will only be reviewed by the Academic Editor in the next round.

We look forward to receiving your revised manuscript.

Kind regards,

Baogui Xin, Ph.D.

Academic Editor

PLOS ONE

Journal Requirements:

Reviewers' comments:

Reviewer's Responses to Questions

**Comments to the Author**

1. If the authors have adequately addressed your comments raised in a previous round of review and you feel that this manuscript is now acceptable for publication, you may indicate that here to bypass the “Comments to the Author” section, enter your conflict of interest statement in the “Confidential to Editor” section, and submit your "Accept" recommendation.

Reviewer #1: (No Response)

Reviewer #2: All comments have been addressed

2. Is the manuscript technically sound, and do the data support the conclusions?

Reviewer #1: Yes

Reviewer #2: Yes

3. Has the statistical analysis been performed appropriately and rigorously? 

Reviewer #1: N/A

Reviewer #2: Yes

4. Have the authors made all data underlying the findings in their manuscript fully available?

Reviewer #1: No

Reviewer #2: Yes

5. Is the manuscript presented in an intelligible fashion and written in standard English?

Reviewer #1: Yes

Reviewer #2: Yes

6. Review Comments to the Author

Reviewer #1: General comment –I appreciate the effort you have put into revising and improving your manuscript titled "Exploring the path to corruption – An informed grounded theory study on the decision-making process underlying corruption." The revisions you have made in response to the previous round of review have indeed enhanced the overall quality of your work.

That said, upon careful consideration of this current version, I still see some areas that could use a bit more refinement. These are minor issues and mainly pertain to details that would further improve the clarity and completeness of your study. I am confident that you will be able to address these issues swiftly, as they do not fundamentally alter the scientific content of your work. I am looking forward to receiving your revised manuscript and to seeing this important piece of research ready for publication.

Comment 1 – I've reviewed the updated data availability statement as well as your responses to both me and the editor. I understand your concerns about participant confidentiality in the context of your research on corruption, which indeed deals with sensitive issues. This is an important consideration and certainly a challenge when dealing with such a delicate topic.

However, the principles of data availability are central to the progress of science, which include transparency, verification of results, fostering new research, economic responsibility, skill development, and public trust. These principles apply to all fields of research and play a crucial role in maintaining the credibility and reproducibility of scientific work.

While I fully understand your point of view regarding the interview transcripts, perhaps there is an alternative solution to consider. If providing anonymised transcripts of interviews is not feasible due to concerns over possible identification of participants, perhaps you could consider sharing your field notes, if these are available. They could be edited and anonymised to a greater extent than interview transcriptions and may provide sufficient data to allow replication of your findings without breaching confidentiality agreements with your participants.

I would like to emphasize that this suggestion is in line with maintaining the balance between respecting participant confidentiality and upholding the standards of data availability in scientific research.

Comment 2 – While I appreciate the authors' attempt to explain the rationale for including Indonesian informants in the study, I believe the language used in lines 305-308 on page 13 may inadvertently stigmatise a nation. The statement "Indonesia can be considered a corrupt country" is a potentially stigmatising label that may paint an unbalanced picture of the situation in Indonesia. I understand the authors’ intent to emphasize the prevalence of reported corruption cases in Indonesia as an explanation for their choice of sample; however, this framing could be perceived as negative or offensive.

Corruption is a global issue and exists in varying degrees in every country. Using an index score to label an entire country as "corrupt" oversimplifies a complex issue. It would be more accurate and sensitive to refer to Indonesia's corruption perception index (CPI) scores, indicating a high incidence of reported corruption cases, rather than branding the entire nation as "corrupt."

As an alternative, the authors could consider using a more neutral language, such as "Due to the consistently low Corruption Perception Index (CPI) scores and frequent media reports of corruption cases at various levels of government, Indonesia was selected as the focal context for this study." This rephrasing provides the necessary justification for the authors' sample choice while avoiding potentially harmful stereotyping.

Reviewer #2: The authors have adequately addressed comments raised in a previous round of review. In the manuscript，authors explored new forms of corrupt decision-making elements within this process, and discovered some important cognitive processes that lead up to corrupt actions. Although the manuscript has some shortcomings in the innovation of its conclusions, it has some value in the methodology of studying corruption through grounded theory.

7. PLOS authors have the option to publish the peer review history of their article (what does this mean?). If published, this will include your full peer review and any attached files.

Reviewer #1: No

Reviewer #2: No

---

## [Author Response · Author response to Decision Letter 1]

4 Sep 2023

Dear Prof. Baogui Xin,

Thank you for giving us the opportunity to address some minor revisions. We have adapted our manuscript based on your and the reviewers’ comments. Below are our responses to your and reviewers’ comments and suggestions. 

Responses to the Comments of the Editor

Response: Thank you for reminding us to check our reference list. We have checked our references once more and can conclude that the reference list is complete and correct. We also checked our references through Retraction Watch Database (http://retractiondatabase.org) and found no retracted articles in our reference list. While we have been thorough in our search, we appreciate your specific guidance if we would have unadvertedly cited a retracted manuscript. 

Responses to the Comments of Reviewer #1

General comment –I appreciate the effort you have put into revising and improving your manuscript titled "Exploring the path to corruption – An informed grounded theory study on the decision-making process underlying corruption." The revisions you have made in response to the previous round of review have indeed enhanced the overall quality of your work.

That said, upon careful consideration of this current version, I still see some areas that could use a bit more refinement. These are minor issues and mainly pertain to details that would further improve the clarity and completeness of your study. I am confident that you will be able to address these issues swiftly, as they do not fundamentally alter the scientific content of your work. I am looking forward to receiving your revised manuscript and to seeing this important piece of research ready for publication.

Response: Thank you for your positive feedback regarding our revisions. Indeed, we found that your constructive feedback has helped us a lot to improve the quality of our manuscript. We would also like to thank you for providing some additional minor comments on this current version. Please find our responses to your current comments below.

Comment 1 – I've reviewed the updated data availability statement as well as your responses to both me and the editor. I understand your concerns about participant confidentiality in the context of your research on corruption, which indeed deals with sensitive issues. This is an important consideration and certainly a challenge when dealing with such a delicate topic.

However, the principles of data availability are central to the progress of science, which include transparency, verification of results, fostering new research, economic responsibility, skill development, and public trust. These principles apply to all fields of research and play a crucial role in maintaining the credibility and reproducibility of scientific work.

While I fully understand your point of view regarding the interview transcripts, perhaps there is an alternative solution to consider. If providing anonymised transcripts of interviews is not feasible due to concerns over possible identification of participants, perhaps you could consider sharing your field notes, if these are available. They could be edited and anonymised to a greater extent than interview transcriptions and may provide sufficient data to allow replication of your findings without breaching confidentiality agreements with your participants.

I would like to emphasize that this suggestion is in line with maintaining the balance between respecting participant confidentiality and upholding the standards of data availability in scientific research.

Response: Unfortunately, we did not create any specific “field notes” during our research (please note that this may depend on the qualitative approach taken). We would like to offer to share our coding dictionary in the Supporting Information, however, to adhere to the transparency standards of good research. As we indicated in the manuscript (lines 386-399, p. 17), the interviews were done in the Indonesian language and were coded in original transcripts and audio recordings. Therefore, most of the quotes in the coding dictionary are in the Indonesian language. Some of them were translated into English to let readers who do not speak Indonesian have some impressions about the coding process. 

Comment 2 – While I appreciate the authors' attempt to explain the rationale for including Indonesian informants in the study, I believe the language used in lines 305-308 on page 13 may inadvertently stigmatise a nation. The statement "Indonesia can be considered a corrupt country" is a potentially stigmatising label that may paint an unbalanced picture of the situation in Indonesia. I understand the authors’ intent to emphasize the prevalence of reported corruption cases in Indonesia as an explanation for their choice of sample; however, this framing could be perceived as negative or offensive.

Corruption is a global issue and exists in varying degrees in every country. Using an index score to label an entire country as "corrupt" oversimplifies a complex issue. It would be more accurate and sensitive to refer to Indonesia's corruption perception index (CPI) scores, indicating a high incidence of reported corruption cases, rather than branding the entire nation as "corrupt."

As an alternative, the authors could consider using a more neutral language, such as "Due to the consistently low Corruption Perception Index (CPI) scores and frequent media reports of corruption cases at various levels of government, Indonesia was selected as the focal context for this study." This rephrasing provides the necessary justification for the authors' sample choice while avoiding potentially harmful stereotyping. 

Response: Thank you for noticing that our language in lines 305-308 page 13 potentially stigmatizes the country and could be perceived as negative and offensive. We appreciate the suggested rephrasing sentences to avoid stigmatizing Indonesia as a corrupt country. We have now changed the sentences based on your suggestion (lines 305-307, page 13). 

Responses to the Comments of Reviewer #2

The authors have adequately addressed comments raised in a previous round of review. In the manuscript, authors explored new forms of corrupt decision-making elements within this process, and discovered some important cognitive processes that lead up to corrupt actions. Although the manuscript has some shortcomings in the innovation of its conclusions, it has some value in the methodology of studying corruption through grounded theory.

Response: Thank you for your positive feedback. Your valuable comments have provided important insights for us to enhance our work.

---

## [Editor Report · Decision Letter 2]

7 Sep 2023

Exploring the path to corruption – An informed grounded theory study on the decision-making process underlying corruption

PONE-D-23-05981R2

Dear Dr. Manara,

We’re pleased to inform you that your manuscript has been judged scientifically suitable for publication and will be formally accepted for publication once it meets all outstanding technical requirements.

Kind regards,

Baogui Xin, Ph.D.

Academic Editor

PLOS ONE
---

## [Editor Report · Acceptance letter]

12 Sep 2023

PONE-D-23-05981R2 

Exploring the path to corruption – An informed grounded theory study on the decision-making process underlying corruption 

Dear Dr. Manara:

I'm pleased to inform you that your manuscript has been deemed suitable for publication in PLOS ONE. Congratulations! Your manuscript is now with our production department. 

Kind regards, 

on behalf of

Professor Baogui Xin 

Academic Editor

PLOS ONE